# U0126 Compound Triggers Thermogenic Differentiation in Preadipocytes via ERK-AMPK Signaling Axis

**DOI:** 10.3390/ijms24097987

**Published:** 2023-04-28

**Authors:** Sunday Amos Onikanni, Cheng-Ying Yang, Lloyd Noriega, Chih-Hao Wang

**Affiliations:** 1Graduate Institute of Biomedical Sciences, College of Medicine, China Medical University, Taichung 406040, Taiwan; 2Department of Chemical Sciences, Biochemistry Unit, Afe Babalola University, Ado-Ekiti 360101, Ekiti State, Nigeria; 3Graduate Institute of Cell Biology, College of Life Sciences, China Medical University, Taichung 406040, Taiwan

**Keywords:** U0126, MEK inhibition, adipose tissue, obesity, adipogenesis, thermogenesis, AMPK

## Abstract

In recent years, thermogenic differentiation and activation in brown and white adipose tissues have been regarded as one of the major innovative and promising strategies for the treatment and amelioration of obesity. However, the pharmacological approach towards this process has had limited and insufficient commitments, which presents a greater challenge for obesity treatment. This research evaluates the effects of U0126 compound on the activation of thermogenic differentiation during adipogenesis. The results show that U0126 pretreatment primes both white and brown preadipocytes to upregulate thermogenic and mitochondrial genes as well as enhance functions during the differentiation process. We establish that U0126-mediated thermogenic differentiation induction occurs partially via AMPK activation signaling. The findings of this research suggest U0126 as a promising alternative ligand in pursuit of a pharmacological option to increase thermogenic adipocyte formation and improve energy expenditure. Thus it could pave the way for the discovery of therapeutic drugs for the treatment of obesity and its related complications.

## 1. Introduction

Obesity has become a global epidemic, increasing the risk of developing metabolic disorders, such as type 2 diabetes (T2D), hypertension, dyslipidemia, coronary heart disease, and certain types of cancer [1,2,3,4]. Physical exercise and dietary changes are known strategies for losing weight and managing obesity-related disorders. Calorie restriction (CR) thus serves as an avenue for weight reduction and longer lifespans [5,6]. However, in the last three decades, the prevalence of obesity/overweight in both adults and children has increased worldwide, and approximately 40% of the world’s adult population is currently overweight or obese [2,7]. The rate of obesity and overweight individuals in the United States has more than double between 1976–1980 and 2013–2014, revealing that two-thirds of American adults are now overweight or obese [8,9]. Furthermore, obesity depletes a significant number of societal resources. It is estimated, statistically, that by 2030, medical expenditures related to overweight and obesity in the United States will amount USD 861–957 billion, accounting for 16–18% of the country’s overall healthcare costs [10,11].

It has been little over a decade since metabolically active brown adipose tissue (BAT) in healthy adult humans was definitively identified [12,13]. Importantly, clinical cross-sectional research revealed that BAT activity declines with age, as measured by FDG-PET/CT, coinciding with the development of obesity and insulin resistance [14]. Since then, researchers have discovered that human BAT expresses the thermogenic uncoupling protein 1 (UCP1), as well as its ability to dissipate energy. BAT creates heat by combusting macronutrients, a process known as thermogenesis. In addition to using its thermogenic activity to defend the body against glucose and lipid accumulation, BAT has been identified as an endocrine organ which communicates with other tissues to maintain whole-body energy metabolism, glucose homeostasis, and inflammation [15,16,17]. Thermogenesis, or the thermogenic program, is triggered by cold sensations or food intake. Sympathetic neurons produce norepinephrine, which activates β3-adrenergic receptors in BAT and white adipose tissue (WAT), leading to the upregulation of mitochondrial genes and thermogenic genes, such as UCP1, via the cAMP-mediated activation of the protein kinase A (PKA) and p38 MAP kinase (p38 MAPK) axis [18,19]. In addition, AMPK has been shown to be one of the induction signals for UCP1 expression via PGC1α, resulting in the thermogenic differentiation of WAT and the activation of BAT [20,21]. The bone morphogenic protein 7 (BMP7) interacts with a heterotetrameric complex, consisting of two type 1 and two type 2 transmembrane serine/threonine kinase receptors, and functions as an autocrine/paracrine agent in mice to facilitate classical brown adipocyte differentiation. BMP7 promotes brown preadipocyte differentiation even in the absence of adipogenic induction cocktails. By activating the p38 MAPK pathway, BMP7 elevates the expression of UCP1, PGC1α, and PRDM16 [22]. BMP7 also upregulates UCP1 expression in the selected clones of immortalized white and brown preadipocytes derived from the human neck [23]. Therefore, with the reality that obesity is recognized as one of the leading causes of death in developed countries, an urgent attention to ameliorate this issue is thus required.

Indeed, cold exposure and adrenomimetic drugs have been shown to significantly increase thermogenic differentiation and activation in BAT and WAT, leading to the consequent energy expenditure in humans through extensive investigations [24,25,26]. However, due to potential detrimental side effects on cardiovascular function, treatment strategies aiming at a general activation of the sympathetic nervous system and adrenergic receptors may not be appropriate for obese and diabetic patients [14]. Thus, the identification of more potent and specific pharmacological products targeting thermogenic activation in WAT and BAT with little or no adverse effects is needed.

Connectivity Map (CMAP) is a publicly available database and software tool. It is a large-scale resource that contains the gene expression profiles of human cells treated with various small molecules, such as drugs or other compounds. The purpose of CMAP is to provide a comprehensive resource for studying the molecular mechanisms of drugs and other bioactive compounds and to enable the discovery of new therapeutic targets and drug candidates. The CMAP database contains gene expression data for over 7000 small molecules and more than 1.3 million gene expression profiles across various human cell lines. The CMAP software tool allows users to search the database for small molecules that produce similar gene expression changes, based on user-supplied query gene sets. One of the key applications of CMAP is to identify potential drug targets and drug candidates by identifying compounds that induce similar gene expression changes to those observed in disease states. By comparing the gene expression profiles of cells treated with different compounds, researchers can identify common molecular pathways and mechanisms of action and potentially identify new targets for drug development [27].

U0126 is a synthetic small molecule inhibitor that targets the mitogen-activated protein kinase kinase (MAPKK/MEK) family of enzymes. U0126 inhibits MEK1 and MEK2 by binding to their ATP-binding sites, thereby blocking the phosphorylation and activation of MAPKs, such as extracellular signal-regulated kinases (ERK1/2) [28]. U0126 is widely used as a pharmacological tool in both basic and clinical research to study the role of MAPK signaling in various cellular processes and disease models. U0126 has been used to investigate the role of MAPK signaling in cancer cell proliferation, migration, and invasion and to evaluate the efficacy of MEK inhibitors as potential anticancer drugs. U0126 has also been used to study the role of MAPK signaling in other physiological and pathological conditions, such as inflammation, neurodegeneration, and cardiovascular disease [29].

In this study, we utilized CMAP to investigate a new strategy for obesity treatment that focuses on enhancing energy expenditure through the activation of brown adipocytes and browning of white adipocytes. We found that U0126, a potent MEK 1/2 inhibitor, can induce a thermogenic differentiation program in human white and brown preadipocytes, resulting in increased expression of UCP1, mitochondrial biogenesis, and thermogenic function. The use of U0126 could be a cost-effective and safe approach to target obesity. This study therefore suggests that the repurposing of existing drugs, such as U0126, to target existing challenges in the field could minimize side effects and preserve their existing value.

## 2. Results

### 2.1. Bioinformatic Strategy Identified Small Molecules for Priming Thermogenic Differentiation in Human Progenitor Cells

Previously, Xue and colleague isolated clonal preadipocytes from human white and brown fat and identified the gene signature that could be used for prediction of thermogenic differentiation potential in human preadipocytes [23]. To identify the compounds that were able to activate thermogenic differentiation potential in preadipocytes, we aimed for the compounds that potentially trigger similar gene signatures, as in a previous study [23]. We selected genes that were most positively and negatively correlated with UCP1 expression [23] as a thermogenic differentiation signature for Connectivity Map (CMAP) analysis (http://clue.io/cmap (accessed on 15 September 2021)). The CMAP database utilizes pattern-matching algorithms to compare input gene signatures with gene signatures in diverse contexts of over 450,000 chemical compounds. The compounds were ranked depending on their similarity or dissimilarity score in regard to the input thermogenic differentiation gene signatures (Appendix A). Among the top-ranking candidates, we selected U0126 compound (Figure 1A) to evaluate whether it could induce both human white and brown preadipocytes to acquire or enhance thermogenic differentiation potential.

### 2.2. U0126 Pretreatment Drives the Thermogenic Differentiation of Human White Preadipocytes

To investigate whether U0126 treatment can prime human progenitor cells to acquire gene signature for thermogenic differentiation, we pretreated human white preadipocytes, which were isolated from human superficial neck fat, with different concentrations (0, 0.1, 0.3, 1, 3, and 10 μM) of U0126 for 6 days and then induced them to adipogenic differentiation. First, a 6-day U0126 pretreatment caused no obvious cytotoxicity at any of the concentrations tested (Figure 1B). Most importantly, the expression of thermogenic genes, including UCP1 and iodothyronine deiodinase 2 (DIO2), was increased in white preadipocytes pretreated with U0126 in a dose-dependent manner (Figure 1C,D). The elevation of mitochondrial biogenesis and function is a hallmark of thermogenic differentiation and function [30,31]. U0126 pretreatment also stimulated preadipocytes to upregulate the expression of mitochondrial biogenesis regulators, such as peroxisome proliferator-activated receptor gamma coactivator 1α (PGC1α) and nuclear respiratory factor 1 (NRF1) (Figure 1E,F and Appendix A).

BMP7 pretreatment is a well-known stimulus for priming thermogenic differentiation in preadipocytes [23,32]. Therefore, following the previous findings, we incorporated a positive control group into the experiment, which received 3.3 nM BMP7 of pretreatment. After comparison with the BMP7 group, we discovered that a low dose of U0126 (0.3 μM) revealed a comparable effect on the reporter assay of UCP1 transcriptional activity, similarly to the BMP7 pretreatment (Figure 1G and Appendix A). Intriguingly, a high dose of U0126 (3 μM) stimulated much better thermogenic differentiation potential in human white preadipocytes (Figure 1G and Appendix A). Moreover, U0126 pretreatment did not change the morphology of the lipid formation monitored by Oil Red O staining (Figure 2A–C) and the expression of general adipogenic markers, such as fatty acid binding protein 4 (AP2) and fatty acid synthase (FAS) (Figure 2D,E and Appendix A).

To prove whether genetic alterations indeed contribute to changes in protein expression and function, we examined PGC1α, NRF1 and UCP1 protein levels, and thermogenesis capacity in differentiated white adipocytes with BMP7 or U0126 pretreatment. In white adipogenesis, U0126 pretreatment upregulated the protein levels of PGC1α and NRF1 (Appendix A) and UCP1 protein levels (Figure 3A,B) and granted differentiated adipocytes with a high capacity of thermogenesis function (Figure 3C,D), which was monitored by ERthermAC dye [33,34], compared to the DMSO control group. Notably, a high dose of U0126 promoted a much greater thermogenesis function during white preadipocytes differentiation compared to BMP7 pretreatment (Figure 3C,D). Taken together, these findings demonstrate that U0126 can confer white preadipocytes with the potential for thermogenic differentiation.

### 2.3. U0126 Pretreatment Confers Human Brown Preadipocytes to Acquire Enhanced Mitochondrial and Thermogenic Function after Differentiation

To understand whether U0126 could strengthen the thermogenic program and make brown adipocytes more active after differentiation, we examined the U0126 pretreatment effect on human brown adipocytes compared to DMSO and 3.3 nM BMP7 pretreatment. U0126 did not have any effect on the cell viability of human brown preadipocytes at a concentration of less than 3 μM but increased cell proliferation at a 10 μM dosage (Figure 4A and Appendix A). Compared to the DMSO group, although there were decreases in the AP2 level of U-0.1 group and the FAS level of U-0.3 group, there were no obvious changes to the expression of AP2 and FAS or the lipid droplet formation in other U0126 pretreatment groups (Figure 4B–D and Appendix A). This indicated that U0126 pretreatment did not alter general adipogenic differentiation in human brown preadipocytes. However, thermogenic genes (UCP1 and DIO2) and mitochondrial biogenesis-related genes (PGC1α and NRF1) were dose-dependently upregulated after differentiation in the U0126 pretreatment group (Figure 5A–D). In addition, those differentiated brown adipocytes displayed a higher mitochondrial oxygen consumption rate (Figure 5E,F) and thermogenic function (Figure 5G,H) in response to forskolin stimulation. Compared to the BMP7 pretreatment group, U0126 triggered either comparable or higher inductions in the thermogenic gene expression and function (Figure 5E–H and Appendix A). This suggests that U0126 pretreatment may stimulate brown preadipocytes to become active and functional brown adipocytes after adipogenesis.

### 2.4. U0126-Mediated AMPK Phosphorylation Involves Thermogenic Activation in Human White Adipogenesis

To unravel the mechanisms underlying U0126-mediated thermogenic differentiation in white adipogenesis, we dissected the signaling pathways in U0126-pretreated cells. U0126 is a well-known MEK1/2 inhibitor [29,35]. It has been shown that the inhibition of ERK1/2, the downstream target of MEK1/2, promotes AMPK and p38 activations in different cell types [36,37,38,39]. Given that AMPK and p38 are two major crucial activators in thermogenic signaling pathways, we hypothesized that U0126-mediated thermogenic differentiation may result from AMPK or p38 activation through ERK inhibition. Thus, we measured the phosphorylation of ERK1/2 and AMPK in human white preadipocytes pretreated with or without 3 μM of U0126. We observed that ERK1/2 phosphorylation was inhibited (Figure 6A,B) and AMPK phosphorylation was significantly activated (Figure 6C,D) in white preadipocytes with U0126 treatment. To demonstrate the importance of AMPK activation, we co-treated cells with a selective AMPK inhibitor, Compound C. The results revealed that U0126-mediated thermogenic differentiation activation, such as the induction of UCP1, DIO2, and PGC1α genes, were partially inhibited via the co-treatment of Compound C (Figure 6E–G). These findings together suggest that U0126-mediated AMPK activation plays an essential role for priming white preadipocytes to acquire thermogenic differentiation potential.

## 3. Discussion

The development of obesity has been identified as being due to an imbalance between energy intake and expenditure. Previous anti-obesity strategies have focused on energy intake inhibition, such as appetite suppressants and lipid absorption inhibitors [40,41]. New strategies for obesity treatment center on the enhancement of energy expenditure via the browning of white adipocytes and the activation of brown adipocytes that has been proven to expend energy by heat production through UCP1 [42]. Our data demonstrates that U0126 pretreatment induced a thermogenic differentiation program, such as UCP1 expression, mitochondrial biogenesis, and thermogenic function, during white preadipocyte differentiation. Other studies revealed the high level of UCP1 in brown adipocytes under basal conditions, while the expression is induced apparently in response to a few thermogenic stimuli or agonists [43]. In this study, we demonstrated that U0126 is a potent stimulus that can further enhance the thermogenic system in human brown adipocytes.

Since therapeutic agents devoid of side effects could be the way forward, there is a need to develop a cost-effective drug that is easier to produce. One such approach is the use of alternative systems of medicine, which could re-propose existing drugs to target existing challenges, while minimizing side effects and preserving their existing value [44]. U0126 is known as a potent, non-ATP competitor and selective MEK 1/2 inhibitor, with inhibitory effects on autophagy and mitophagy [45,46]. U0126 treatment has been shown to reduce the growth of different tumors in vitro and in vivo [45,47]. Studies have revealed its ability to activate vasoconstrictor receptors in male rats by inhibiting the MEK-ERK1/2 pathway both in stroke and organ culture models [46]. In the present study, we examined another role of U0126 in modulating thermogenic differentiation in human white and brown preadipocytes via ERK inhibition, followed by the activation of the AMPK signaling axis. Recently, a mouse study that applied thermogenic gene signatures from BAT and cold-induced WAT to CMAP analysis also identified U0126 as one of the top-ranking candidates [48]. Obese mice displayed a high level of ERK phosphorylation in WAT compared to lean mice. Consistently, the daily administration of MEK inhibitors, including GSK1120212 or PD0325901, for a week upregulated thermogenic genes in WAT and improved glucose homeostasis in obese mice [49]. Furthermore, U0126 treatment was found to reverse the inhibitory effect of central resistin on thermogenesis [50], and U0126 was also shown to regulate thermogenesis induced by Baicalin, a flavone compound extracted from plant roots, thus potentially counteracting diet-induced obesity [51]. Prior research has failed to extensively explore relevant mechanisms, but our study sheds light on the potential underlying actions of U0126 in the thermogenic differentiation of both white and brown adipocytes.

## 4. Materials and Methods

### 4.1. Identification of Small Molecules as Potential Thermogenic Activators

We selected the top 100 genes that were most positively and negatively correlated with UCP1 expressions from the previous study [23] as a thermogenic differentiation signature for Connectivity Map (CMAP) analysis (http://clue.io/cmap (accessed on 15 September 2021)). The CMAP database utilizes pattern-matching algorithms to compare input gene signatures with gene signatures in diverse contexts of over 450,000 chemical compounds. The CMAP-generated enrichment scores for these compounds were ranked depending on their similarities to the input thermogenic differentiation gene signatures. The top 10 candidate compounds with an enrichment score of over 70 are listed in Appendix A.

### 4.2. Cell Culture

The immortalized human white and brown preadipocytes were maintained in (DMEM, GIBCO) with 10% fetal bovine serum (FBS, Millipore Sigma, Burlington, MA, USA) in 5% CO_2_ and 37 °C conditions. Initially, cells were pre-treated with BMP7 or U0126 for 6 days to induce thermogenic differentiation commitment. In the next step, the cells were induced to differentiate by induction medium, i.e., DMEM containing 10% FBS, 0.5 mM 3-isobutyl-1-methylxanthine, 0.1 μM dexamethasone, 17 μM pantothenic acid, 3.3 μM biotin, 2 nM T_3_, 31.25 μM Indomethacin, and 3 μg/mL insulin, for 12 days. The induction media was changed every 3 days. After these procedures, the mature adipocytes were maintained in DMEM containing 10% FBS.

### 4.3. Viability Assay

The cell viability of the treated preadipocytes was assessed according to the method described by Lall et al. [52]. The preadipocytes were seeded in a 96-well plate. After the treatment of U0126 for 6 days, the cells were incubated with 10 μL of presto reagent and 90 µL of DMEM. The incubation was in 5% CO_2_ and at 37 °C and covered with aluminum foil to avoid light interference. After 20 min, the mixture was collected and was read at a fluorescence of 560 nm by a spectrophotometer.

### 4.4. RNA Extraction and Gene Expression Analysis

The adipocytes were lysed, and total RNA was purified using TRIzol reagent. Reverse transcription was performed with RNA to obtain cDNA using a First Strand cDNA Synthesis Kit. The expression level of specific genes was analyzed by a quantitative real-time PCR system using a SYBRgreen PCR mix. h18S was further used as a housekeeping gene for normalization. The primer sequences for amplifying specific genes are listed in Appendix A.

### 4.5. UCP1 Reporter Assay

The white preadipocytes were transfected with a construct containing a luciferase gene driven by human UCP1 promoter. After differentiation, mature adipocytes were lysed using the Luci-150 Gen assay reagent (InvivoGen, San Diego, CA, USA). After 10 min of incubation, 20 µL lysate was transferred to a new 96-well microplate and luciferase activity was measured.

### 4.6. Oil Red O Staining

The Oil Red O staining solution was prepared by 3 parts of Oil Red O (0.5% (*v*/*v*) in isopropanol) mixed with 2 parts of water and then filtered through a 0.45 mm filter. Mature adipocytes were washed 3 times with PBS and fixed for 30 min with 4% (*v*/*v*) paraformaldehyde (PFA). The fixed cells were incubated with filtered Oil Red O staining solution for 1 h or more at room temperature. For quantification, Oil Red O inside the cells was eluted by 100% isopropanol and determined the absorbance at a wavelength of 500 nm on a spectrophotometer.

### 4.7. Immunoblotting Assay

Adipocytes lysate was extracted with an RIPA buffer. A total of 30 μg protein was loaded into a 10% SDS-PAGE gel and transferred onto a polyvinylidene difluoride membrane (pore size: 0.22 mm, Merck Millipore, Burlington, MA, USA). The membranes were blocked using a blocking buffer. Primary antibodies were diluted with PBST (1:1000) and added into membranes for overnight incubation to detect specific protein. The secondary antibodies (Jackson, 1:50,000) were then added onto the membrane for one hour. Immunoreactivities were detected with enhanced chemiluminescent autoradiography. The primary antibodies used are listed in Appendix A.

### 4.8. Thermogenesis Assay

Differentiated adipocytes were incubated in FBS-free DMEM mixed with 250 nM thermosensitive fluorescent ERthermAC dye for 30–45 min at 37 °C [34]. Media were replaced with fresh 90 μL DMEM/H without phenol red prior to imaging. Fluorescence in the stained cells was detected via a GloMax Discover Multimode Detection System (Promega, Madison, WI, USA) using 520 nm excitation and emission at 580–640 nm. The temperature inside the machine was equilibrated at 25 °C. After measuring 3 points of basal fluorescence, 10 μL forskolin (final concentration: 10 μM; F6886, Millipore Sigma, Burlington, MA, USA) was added to initiate thermogenesis. Fluorescence was recorded every 5 min over 120 min. Results were interpreted as relative intensity (normalized to basal measurements). The fluorescent intensity of the ERthermAC dye is inversely correlated with intracellular temperature. The area above the curve was quantified to show the thermogenic capacity in response to forskolin.

### 4.9. Oxygen Consumption Rate

The Seahorse XF24 Flux analyzer (Agilent Technologies, Santa Clara, CA, USA) was utilized to measure Oxygen Consumption Rate (OCR) according to the manufacturer’s protocol. Preadipocytes were grown and differentiated into XF24 cell culture microplates. To determine the glucose-dependent OCR, cells were incubated with serum-free DMEM with 25 mM glucose and 1 mM pyruvate and then treated with different drugs to measure different parameters: 2 µM oligomycin (ATP production respiration), 10 µM forskolin (FSK-dependent respiration), and 2 µM antimycin A (non-mitochondrial respiration) [34].

### 4.10. Quantification and Statistical Analysis

All statistics were calculated using Microsoft Excel and GraphPad Prism. A two-tailed Student’s *t*-test was performed for all two group comparisons. A one-way ANOVA followed by a Tukey’s post hoc test was performed when comparing more than three groups. Each datum was presented as the mean ± SEM. Significance was defined as * *p* < 0.05, ** *p* < 0.01, *** *p* < 0.001, and **** *p* < 0.0001.

## Figures and Tables

**Figure 1 ijms-24-07987-f001:**
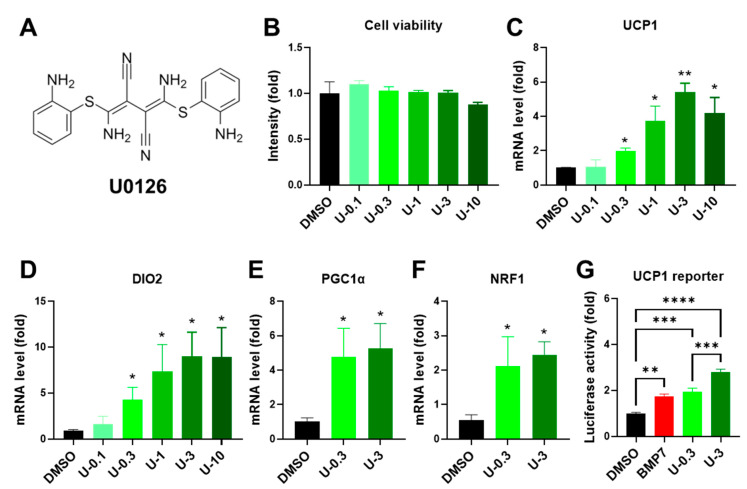
U0126 pretreatment upregulates thermogenic genes after the adipogenesis of human white preadipocytes. (**A**) The structure of U0126 compound. (**B**) Human white preadipocytes were pretreated with different concentrations (0, 0.1, 0.3, 1, 3, and 10 μM) of U0126 (DMSO, U-0.1, U-0.3, U-1, U-3, and U-10) for 6 days. The cell viability was measured. (**C**–**F**) After U0126 pretreatment, preadipocytes underwent adipogenic differentiation for 12 days. Thermogenic genes, UCP1 (**C**) and DIO2 (**D**), and mitochondrial biogenesis genes, PGC1α (**E**) and NRF1 (**F**), were determined. (**G**) The transcriptional activity of the UCP1 gene was monitored by reporter assay. The concentration of BMP7 was 3.3 nM. Unpaired Student’s *t*-tests were compared to the DMSO group. * *p* < 0.05, ** *p* < 0.01, *** *p* < 0.001, **** *p* < 0.0001.

**Figure 2 ijms-24-07987-f002:**
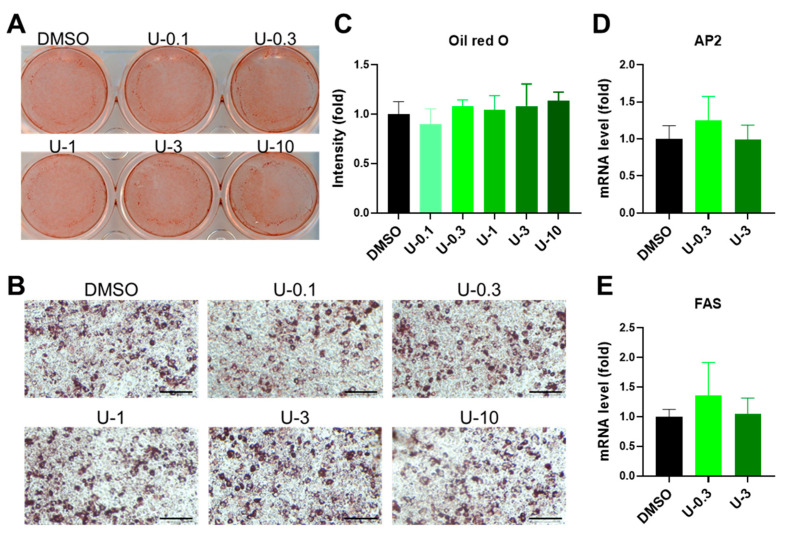
U0126 pretreatment did not affect general adipogenic differentiation in human white preadipocytes. Human white preadipocytes were pretreated with different concentrations (0, 0.1, 0.3, 1, 3, and 10 μM) of U0126 (DMSO, U-0.1, U-0.3, U-1, U-3, and U-10) for 6 days and underwent adipogenic differentiation for 12 days. (**A**–**C**) Mature adipocytes were stained by Oil Red O. Gross view of wells (**A**), images of differentiated adipocytes (**B**), and the quantification of Oil Red O intensity after elution (**C**) were shown. Scale bar, 600 μm. (**D**,**E**) General adipogenic marker genes, AP2 (**D**) and FAS (**E**), were measured in mature white adipocytes. Unpaired Student’s *t*-tests were compared to the DMSO group.

**Figure 3 ijms-24-07987-f003:**
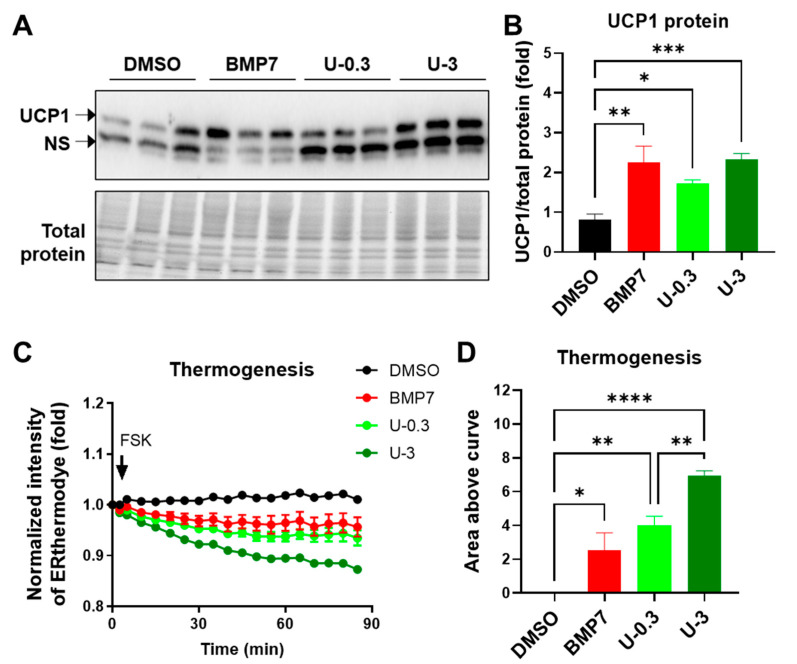
U0126 pretreatment promotes thermogenic function after the adipogenesis of human white preadipocytes. Human white preadipocytes were pretreated with 3.3 nM of BMP7 or 0.3 and 3 μM of U0126 (U-0.3 or U-3) for 6 days and underwent adipogenic differentiation for 12 days. (**A**,**B**) Blot images of UCP1 protein (**A**) and the quantification of band intensity (**B**) in differentiated white adipocytes. NS, non-specific band. (**C**,**D**) Thermogenesis function of adipocytes was monitored by ERthermAC dye staining. (**C**) The changes of ERthermAC intensity after forskolin stimulation. (**D**) The quantification of area above curve in (**C**). Unpaired Student’s *t*-tests. * *p* < 0.05, ** *p* < 0.01, *** *p* < 0.001, **** *p* < 0.0001.

**Figure 4 ijms-24-07987-f004:**
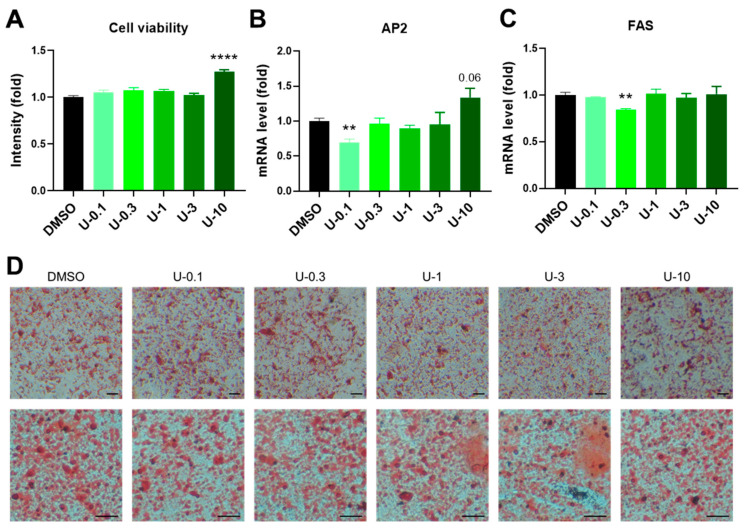
U0126 pretreatment did not alter general adipogenic differentiation in human brown preadipocytes. (**A**) Human brown preadipocytes were pretreated with different concentrations (0, 0.1, 0.3, 1, 3, and 10 μM) of U0126 (DMSO, U-0.1, U-0.3, U-1, U-3, and U-10) for 6 days. The cell viability was measured. (**B**–**D**) After U0126 pretreatment, preadipocytes underwent adipogenic differentiation for 12 days. General adipogenic marker genes, AP2 (**B**) and FAS (**C**), were measured in mature brown adipocytes. (**D**) Images of differentiated brown adipocytes after staining with Oil Red O. Images with low (upper panel) and high (lower panel) magnifications. Scale bar, 600 μm. Unpaired Student’s *t*-tests compared to the DMSO group. ** *p* < 0.01, **** *p* < 0.0001.

**Figure 5 ijms-24-07987-f005:**
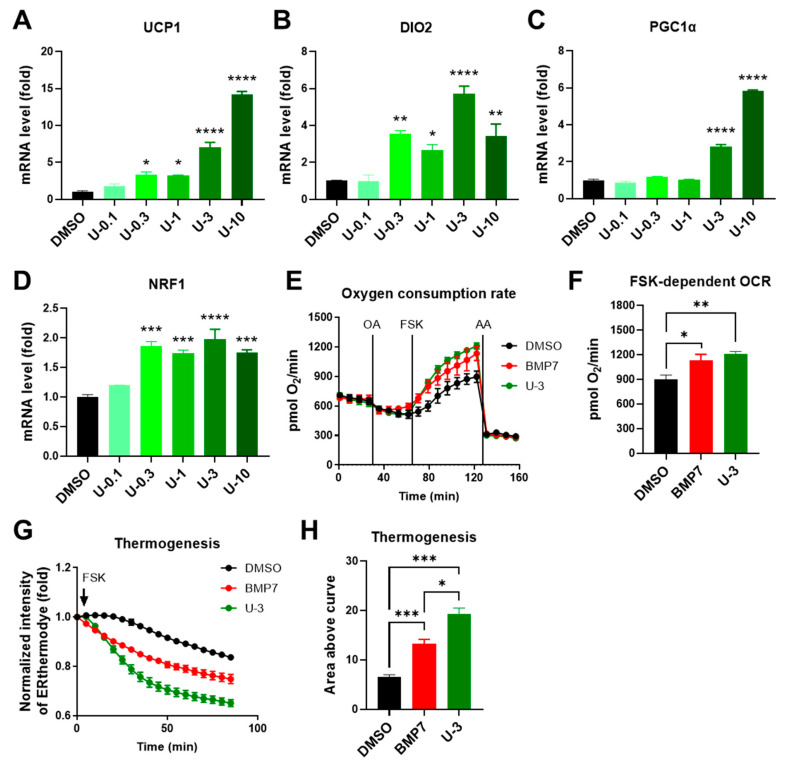
U0126 pretreatment promotes thermogenic function after the adipogenic differentiation of human brown preadipocytes. Human brown preadipocytes were pretreated with different concentrations (0, 0.1, 0.3, 1, 3, and 10 μM) of U0126 (DMSO, U-0.1, U-0.3, U-1, U-3, and U-10) or BMP7 (3.3 nM) for 6 days and underwent adipogenic differentiation for 12 days. (**A**–**D**) Thermogenic genes, UCP1 (**A**) and DIO2 (**B**), and mitochondrial biogenesis genes, PGC1α (**C**) and NRF1 (**D**), were determined. (**E**,**F**) The oxygen consumption rate (OCR; **E**) of differentiated brown adipocytes was monitored while adding different drugs (OA, oligomycin A; FSK, forskolin; AA, antimycin A). (**F**) The quantification of maximal forskolin-induced OCR. (**G**,**H**) The thermogenesis function of adipocytes was monitored through ERthermAC dye staining. (**G**) The changes of ERthermAC intensity after forskolin stimulation. (**H**) The quantification of the area above curve in (**G**). Unpaired Student’s *t*-tests were compared to the DMSO group. * *p* < 0.05, ** *p* < 0.01, *** *p* < 0.001, **** *p* < 0.0001.

**Figure 6 ijms-24-07987-f006:**
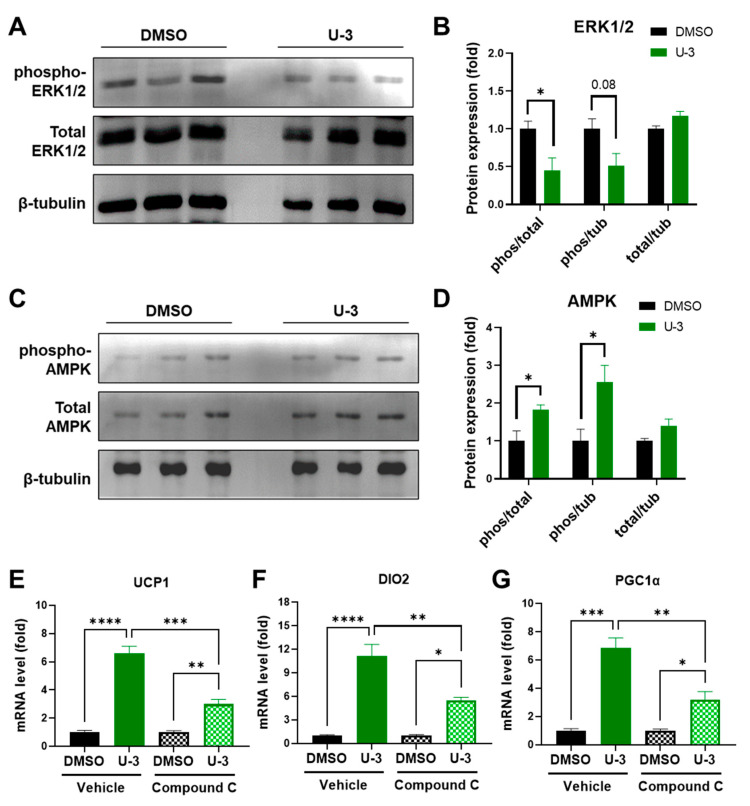
U0126-mediated AMPK activation through MEK inhibition drives the thermogenic differentiation of human white preadipocytes. (**A**–**D**) Human white preadipocytes were pretreated with 3 μM of U0126 (U-3) or without (DMSO) for 6 days. Phosphorylation and total level of ERK1/2 (**A**) and AMPK (**C**) were detected by immunoblotting. (**B**,**D**) The quantification of band intensity in (**A**,**C**), respectively. Ratios between phosphorylated and total protein (phos/total) and phosphorylated protein and β-tubulin (phos/tub), as well as total protein and β-tubulin (total/tub), were shown. (**E**–**G**) Human white preadipocytes were co-pretreated with 3 μM of U0126 (U-3) and an AMPK inhibitor, 10 μM of Compound C, for 6 days and underwent adipogenic differentiation for 12 days. The expression of UCP1 (**E**), DIO2 (**F**), and PGC1α (**G**) were determined. Unpaired Student’s *t*-tests were compared to the DMSO group. * *p* < 0.05, ** *p* < 0.01, *** *p* < 0.001, **** *p* < 0.0001.

## Data Availability

Not applicable.

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
