# Peer review of "U0126 Compound Triggers Thermogenic Differentiation in Preadipocytes via ERK-AMPK Signaling Axis"

_ijms, 2023, doi:10.3390/ijms24097987_

Round 1

Reviewer 1 Report

I have a few comments and questions regarding the manuscript “U0126 Compound Triggers Thermogenic Differentiation in Preadipocytes via ERK-AMPK Signaling Axis”.

First, the introduction needs improvement. More background of this research should be provided, including the method of discovery the compound and the compound U0126. The control compound BMP7 should also be talked about in the introduction.

In Figure 1, experiments were performed with U0126 of different concentrations. However, the author only showed results of mRNA expression in Figure 1E,F and results in 1G with concentrations of 0.3 and 3. The author should show the results of all the concentrations, which can be more conclusive. Same for Figure 2 and the rest. At least, include them in the supplementary.

The author should specify the concentration of control compound BMP7 in all the figures that used the compound. The control compound used was listed as 3.3 nM while U0126 used were in uM concentrations in Figure 3. The difference in thermogenesis of U0126 and BMP7 could be the concentration difference, which was as big as almost 1000 times. Why a way lower concentration has been selected? What will happen if BMP7 was used at similar level? Please provide comments or additional experiments. This will affect the conclusion greatly, since U0126 was proposed as a promising compound similar to BMP7.

The author should show the results of compound selection results discussed in 4.1, including the discovery of U0126 and other top candidates too.

The author demonstrated huge differences of mRNA expression of UCP1 in Figure 1C, but smaller differences of protein expression in Figure 3B. Are the PCR results convincing? Especially, the author talked about upregulations of PGC1a and NFR based off PCR results. Western blot results of protein expression should be included at least as another way to ensure the conclusion is right.

Line 144 The author claimed “there are no significant changes on expression of AP2 and FAS as well as lipid droplet… (Figure 4B-D)”. This is potentially a false statement. Figure 4B has shown potentially significant differences between U0.1, U10, DMSO. The author should provide comment on this and maybe show the significance analysis of this figure as well. Figure 4C, U0.3 seems to have significant differences from the other groups as well.

It is also strange why BMP7 was not included as a control in experiments discussed in Figure 4~6, since BMP7 has been seen with similar properties. The author should address this or provide with extra experiments.

Something minor, the author should have spell and grammar check.

Line 11, it should be “have”

Line 15, “The results show that U0126 pretreatment primes the both”, “the” should be deleted.

Reviewer 2 Report

      Obesity is already a global epidemic which was found to be associated with multiple types of metabolic disorders like type 2 diabetes, coronary heart disease, hypertension and some types of cancer. Besides of strategies focusing on inhibiting energy intake, enhancement of energy expenditure via browning of white adipocytes and activation of brown adipocytes are also promising options. In this study, the authors reported U0126 , a selective MEK1/2 inhibitor, as a alternative choice to promote thermogenic adipocyte formation and energy expenditure. There are several issues/questions as shown below:

1)      For identifying potential molecules via Connectivity Map (CMap) analysis, did you find other inhibitors which show similar result with U0126? Could you share the criteria adopted for the selection?

2)      As U0126 was known as MEK1/2 inhibitor, I’m curious about whether other MEK1/2 inhibitors would have the same performance?

3)      In line 86, how did you determined the concentration range of U0126 used for the treatment? Is it similar to the usage in clinical?

4)      In line 104, what is the concentration of BMP7 you used for the treatment?

Reviewer 3 Report

Dear Authors, 

Thank you for your work. I have read your paper with great interest. 

I would like to add that S. Kosari et al., 2013 highlighted the importance of ERK1/2 signalling pathway in the brain in mediating the reduction in BAT thermogenesis induced by resistin. Study showed that role of ERK1/2 inhibitor U0126 alone without resistin did not change temperature of BAT and whole body. Curently reviewed paper showed in details how thermogenic genes (UCP1 and DIO2) and mitochondrial biogenesis –related genes (PGC1α and NRF1) were dose-dependently upregulated after differentiation in U0126 pretreatment group.  Kosari et al. paper did not analyse expression of mentioned above genes but suggested that should be mechanism of regulation and Onikanni et al., showed that mechanism. Previously research on investigating.

There were other paper by H. Li et al (2021)  focused on importance of other plant compound Baicalin in the thermogenetic activity and mentioned importance of U0126 in thermogenic process.

Despite previous studies were looking at thermogenic activity they investigate other compunds in more debt but in association with U0126 so I suggest current study is not big but has interesting outcome in explaining role of U0126 in activity of thermogenic genes in both BAT and WAT adipocytes.  

I would like to ask you to look through whole text and correct some minor misspellings and wording , such as

1.    In the abstract I would change wording in sentence (line 13) “challenging masses”, could you please find another synonymous.

2.    On line 147 misspelling “does-depending. “

Thank you for your work.

Round 2

Reviewer 1 Report

No further questions.